# Anesthetic Management of Patients with Dilated Cardiomyopathy Undergoing Noncardiac Surgery

**DOI:** 10.3390/medicina59091567

**Published:** 2023-08-29

**Authors:** Mengxin Li, Han Huang

**Affiliations:** Department of Anesthesiology and Key Laboratory of Birth Defects and Related Diseases of Women and Children, Ministry of Education, West China Second University Hospital, Sichuan University, Chengdu 610041, China; limengxindoct@163.com

**Keywords:** anesthetic management, dilated cardiomyopathy, noncardiac surgery

## Abstract

Dilated cardiomyopathy (DCM), a primary myocardial disease, is characterized by dilation of the left or both ventricles and systolic dysfunction with or without congestive heart failure. DCM per se is a well-recognized risk factor for sudden cardiac death and poor surgical outcomes following noncardiac surgery. Surgical trauma/stress represents unique challenges for DCM patient management. Unfortunately, there is a big knowledge gap in managing DCM patients undergoing non-cardiac surgery. Therefore, the aim of our review is to provide basic facts and current advances in DCM, as well as a practical guideline to perioperative care providers, for the management of surgical patients with DCM, who are quite rare compared with the general surgical population. This review summarizes recent advances in the medical management of DCM as well as perioperative assessment and management strategies for DCM patients undergoing noncardiac surgery. Optimal surgical outcomes depend on multiple-disciplinary care to minimize perioperative cardiovascular disturbances.

## 1. Introduction

As the most common type of cardiomyopathy, dilated cardiomyopathy (DCM) is characterized by a dilated left ventricle with impaired contractility while excluding other diseases that could cause systolic dysfunction such as hypertension, heart valve diseases, and coronary artery disease. Right ventricular dilation and dysfunction may also be present but are not essential for a diagnosis [1]. DCM can occur at any age. It most commonly occurs in the 30s or 40s and it is relatively rare in children [2]. Advanced age is an independent risk factor for death in patients with DCM [1,3]. It is estimated that DCM affects 1 in 250 people. However, the reported prevalence in DCM is based on screening studies conducted primarily in healthy young adults, and thus may underestimate the true prevalence of this disease as well as the risk of disease-related complications [4]. It is the third most common cause of heart failure worldwide, with an overall poor prognosis, high morbidity and mortality, and a high risk of sudden cardiac death [5,6,7]. Special attention has been paid to DCM, as these patients become the most common candidates for heart transplantation worldwide [8,9]. The etiology of dilated cardiomyopathy is complex and is generally considered to be caused by a combination of genetic and non-genetic factors. It was mentioned that familial DCM accounts for 30 to 50% of all DCM cases [10,11]. The main mode of inheritance of familial dilated cardiomyopathy is autosomal dominant inheritance, but there are also some familial cases showing mitochondrial mutation, autosomal recessive mutation, and X-linked recessive inheritance [12]. Since mutations in many single genes cause DCM, genetic testing usually uses multiple genomes. Tayal et al. [5] proposed the concept of a ‘core disease set’, which accounts for some key mutations in DCM, including the sarcomere genes MYH7 (encoding beta-myosin heavy chains), TNNT2 (encoding troponin T2), TTN (encoding titin), and the LMNA gene, which encodes a nuclear membrane protein. However, non-genetic factors, such as exposure to hazardous substances, diabetes, arrhythmia, and pregnancy, also contribute to the development of DCM in which typical gene mutations are absent [13]. The natural history of dilated heart disease is largely not known. There are two reasons accounting for this. On the one hand, DCM is genetically heterogeneous and can progress at different rates. On the other hand, the insidious onset of dilated heart disease often hinders the accurate diagnosis until it reaches the late stage [14].

## 2. Clinical Features of Dilated Cardiomyopathy

The signs and symptoms of DCM mostly result from the impaired systolic function of the left or both ventricles. Exertional dyspnea and fatigue are the most common symptoms of DCM, due to pump failure of the left ventricle and low cardiac output, and peripheral edema and distention of neck veins due to failure of the right ventricle. As none of the symptoms or signs are DCM-specific, diagnosis depends on history, physical examination, and exclusion of other causes of heart failure, such as hypertension, myocardial infarction, or valvular diseases. Attention should be paid to patients’ familial history of heart disease, heart failure, and sudden cardiac death.

Currently, risk stratification for dilated heart disease mainly relies on the assessment of left ventricular ejection fraction (LVEF), which serves as a key determinant for implantable cardioverter-defibrillators [15]. Although LVEF is an important prognostic factor for dilated cardiomyopathy [16,17], effective risk stratification remains challenging, especially with regard to sudden cardiac death (SCD) [18,19]. The mechanism for SCD in patients with dilated cardiomyopathy remains unclear, but emerging evidence suggests that myocardial fibrosis following ventricular arrhythmias and scar-related reentry may play an important role [20,21]. Therefore, it is now recommended to use late gadolinium enhancement cardiovascular magnetic resonance imaging (LGE-CMR) to identify and quantify myocardial fibrosis [22,23].

DCM is characterized by decreased ejection fraction (EF) and cardiac output (CO). Reduced forward flow leads to increased ventricular end-diastolic volume, ventricular filling pressure, and ultimately, ventricular dilation as a compensation mechanism to maintain cardiac output. This eventually leads to heart failure. It is well known that heart failure, regardless of underlying causes, is associated with a hypercoagulable state due to proinflammatory cytokine production, increased plasma viscosity, and platelet [24]. The combination of a dilated cardiac chamber with decreased cardiac function results in a greater likelihood of coagulation activation and fibrin aggregation, leading to intracardiac thrombosis [25]. It has been suggested that ischemic cardiomyopathy and dilated LV chamber sizes (LVIDD > 60 mm) are independently associated with LV thrombi [26].

Echocardiography remains the primary imaging technique for DCM, which shows cardiac chamber dilation and wall motion abnormalities (hypokinetic movement) [27]. Meanwhile, it can rule out valvular diseases and other abnormalities in the cardiac structure. Abnormalities in electrocardiography are not uncommon in DCM patients, and arrhythmia is often present. The cardiac MRI [28,29,30] is also useful in the diagnosis of DCM, especially in determining the underlying cause. Gene expression profiling and next-generation sequencing methods have also been introduced into the diagnosis of DCM [31,32].

Even with recent advances in the last couple of decades, in many cases, DCM is still life-threatening [27,33]. Survival rates after DCM diagnosis are approximately 69–72% at 1 year and 54–63% at 5 years [34,35]. Sudden death accounts for 30% of all DCM-related deaths [34,36]. Treatments are symptomatic and supportive unless the underlying cause is treatable, such as standard treatment for heart failure with reduced ejection fraction, anticoagulation therapy for atrial fibrillation, and implantable cardioverter-defibrillator for ventricular arrhythmia.

## 3. Preoperative Assessment

Preoperative assessment begins with careful history taking. A recent decline in exercise tolerance indicates deterioration in cardiac pump function. Current medication should be reviewed, and medical therapy for heart failure management should be continued during the perioperative phase unless contraindicated.

DCM-related arrhythmia is initially assessed with electrocardiography (ECG), which should be a mandatory preoperative test in DCM patients undergoing noncardiac surgery. An increase in the resting heart rate suggests an increase in sympathetic nervous system activity and is associated with adverse outcomes [37]. A reduction in the resting HR inpatients with DCM might be a therapeutic option to improve health-related quality of life (Hr-QoL) [38]. Several studies have confirmed that wide QRS is an independent risk factor for mortality and morbidity increases with a longer follow-up extended to 5 years [39,40,41]. Left bundle-branch block (LBBB) is also a risk factor for unfavorable outcomes in patients with CHF caused by DCM. An increased QTe-slope in patients with DCM indicates recent major arrhythmic events, which is considered a useful factor in stratifying the arrhythmic risk [42]. Holter monitoring should be considered for patients reporting symptoms when routine ECG fails to find abnormalities. Sudden death from major ventricular arrhythmias (MVA) remains the leading cause of mortality for idiopathic DCM patients [43,44,45,46,47]. Non-sustained ventricular tachycardia (NSVT) significantly increases the risk of patients with MVA, even in patients with left ventricular ejection fraction (LVEF) larger than 0.35. For these patients, long-term MVA-free survival was as poor as for patients with LVEF < 0.35, regardless of the value of the left ventricular ejection fraction. The new onset of impaired LV dysfunction (not present at the time of diagnosis), higher NYHA functional class, and reduced daily activity tolerance (often suggested by a low mean heart rate upon 24-holter monitoring) are considered the most reliable predictors of MVA [48]. Interestingly, data suggest that the risk of MVA would not be further increased by the presence, frequency, length, and rate of NSVT in patients with more advanced LVEF [48].

Echocardiography is another essential preoperative test for DCM patients, which provides a direct assessment of the systolic and diastolic function of both ventricles. Moreover, the presence of LV thrombus and estimations of left atrial (LA) and LV end-diastolic pressure (LVEDP) can also be obtained with an echocardiograph [49]. LV thrombosis is a common complication of dilated cardiomyopathy [50,51,52], which can also lead to the occurrence of acute cardiovascular events. Therefore, it is necessary to evaluate ultrasound image features of LV thrombus. These include the shape (the thrombus may be mural or protruding in the cavity), motion (the thrombus may be fixed or exhibit different independent motions), and the possible presence of an adjacent left ventricular aneurysm [53]. Pulsed wave-tissue Doppler imaging (PW-TDI) is an echocardiographic technique with high temporal resolution, which was proposed in the early 1990s [54,55]. It is able to discriminate the fine movements of intracardiac masses, and in particular, to identify endocardial vegetations with their characteristic pattern of incoherent motion. Due to an abnormal free oscillation of the structure, its movement speed and direction are independent of the myocardium and surrounding tissues and are not directly correlated with the cardiac cycle [56,57]. Therefore, PW-TDI is useful in detecting ventricular thrombosis and measuring thrombotic mass mobility. Sonaglioni et al. suggested that using PW-TDI to evaluate a mass peak antegrade velocity (Va) of ≥ 10 cm/s may be a novel marker for intracranial thromboembolic events and could rapidly identify hospitalized patients with an increased likelihood of cerebral or systemic embolism at medium-term follow-up [58]. In a case published in 2021, a patient with ischemic dilated cardiomyopathy and biventricular thrombosis, PW-TDI assessment indicated a mass peak Va of < 10 cm/s in both ventricles and there was no embolic event reported during the clinical course despite the patient being in a hypomobility status during this hospitalization [59]. Previous study has shown that a restrictive LV filling pattern is a powerful independent predictor for cardiac mortality in patients with heart failure caused by nonischemic DCM [60]. The restrictive LV filling pattern provides prognostic value that is incremental to other echocardiography-derived variables such as a short IVRT, incoordinate wall motion, and reduced right ventricle long axis function [60]. Increased early filling velocity (E-wave) and short deceleration times (“restricted filling mode”, severe diastolic dysfunction) are associated with severe hemodynamic impairment, late symptoms, and poor prognosis [61]. Right ventricular dysfunction may exist in DCM, which is an important adverse prognostic indicator [62]. Right ventricular dysfunction appears to be related to the degree of left ventricular dysfunction and biventricular involvement, rather than secondary to pulmonary hypertension [63]. A tricuspid annular proximal systolic excursion (TAPSE) < 14 mm can be routinely used to predict poor prognosis [64]. Dobutamine stress echocardiography is widely used to identify induced myocardial ischemia, viability, and scarring in patients with heart failure [65]. The presence of systolic reserve on this test predicts a good response to drugs, mitral valve repair, and other procedures, whereas the absence of systolic reserve predicts a worse survival [66]. The current guidelines recommend ICD as the primary prevention of SCD in patients with DCM with NYHA II to III heart failure and LVEF < 35% [67,68]. Therefore, we may regard LVEF < 35% and NYHA II to III heart failure as predictors for poor prognosis. However, there are four randomized trials indicating that the ICD does not significantly reduce all-cause mortality in patients with DCM and an LVEF < 35% [43,46,69]. During a preoperative assessment for patients with dilated heart disease who have been implanted with ICD, we should clarify the working mode and working status of the implanted ICD [70]. Special attention should be paid to correcting common disorders that would lead to ventricular fibrillation, such as electrolyte disorder, myocardial ischemia, or hypoxemia [71]. The malfunction of an ICD during surgery is mainly caused by electromagnetic interference, mainly caused by electrical surgical instruments. In order to avoid this situation, the options could be reprogramming an ICD by contacting the manufacturer or avoiding the use of electrical surgical instruments [72]. In a case report [73], anesthesiologists temporarily suspended the defibrillating function of an ICD and only used its pacing function during surgery. Bourke et al. further proposed that the automatic defibrillating function of an ICD should be disabled at the time of surgery. Meanwhile, an external cardioversion/defibrillation device should be available [74]. In addition to electrical surgical instruments, care must be taken for other devices that generate pulsed currents, such as peripheral nerve stimulators or evoked potential monitors, as this may lead to ICD failure [74]. A chest X-ray is useful in the assessment of pulmonary congestion, pulmonary edema, pulmonary hypertension, or pleural effusion when indicated. Computed tomography and angiography of coronary arteries are primarily used to rule out ischemic cardiomyopathy. Cardiac magnetic resonance with late gadolinium enhancement and extracellular volume quantification is a useful tool for risk assessment and outcome prediction [29,75].

Level of B-type natriuretic peptide (BNP) is associated with left ventricular end-diastolic pressure, left ventricular wall stress, fibrosis, and systolic dysfunction [76,77]. The serum level of high-sensitivity CRP (hsCRP) is also an independent predictor for survival rates in patients with DCM. Increased concentrations of plasma endothelin (ET) and its precursor (big-ET) in chronic heart failure patients are also significantly associated with clinical outcomes in patients with heart failure [78,79,80,81,82,83,84]. In addition, one study showed that hs-CRP > 3.90 mg/L and NT pro-BNP > 2247 pmol/L were independent markers for all-cause mortality in a large population of patients with DCM. However, big-ET > 0.95 pmol/L was not associated with a higher all-cause mortality [85].

For patients with symptoms of heart failure, even mild anemia can lead to myocardial ischemia. Currently, there is no guideline recommending transfusion threshold in heart failure patients undergoing noncardiac surgery. Therefore, a symptom-guided management approach is recommended [86]. Sougawa H et al. suggested that a lower preoperative hemoglobin level (<12.2 g/dL) was independently associated with a perioperative adverse cardiovascular event (PACE) in patients undergoing noncardiac surgery [87]. Therefore, we should be more aggressive in terms of blood transfusion in DCM patients with low preoperative hemoglobin levels.

DCM patients often require medical management to preserve cardiac function, prevent ventricular remodeling, and improve symptoms. Angiotensin-converting enzyme inhibitors (ACEI), angiotensin receptor blockers (ARB), and β-adrenergic blockers are often prescribed. Generally, these regular medications should be continued in all DCM patients during the perioperative phase. The potential benefits and harms of interrupting regular medical management should be carefully balanced, and current guidelines recommend continuing the use of ACEI on the day of surgery if it is indicated for preventing cardiac remodeling [88]. Cardiac resynchronization therapy (CRT) should be considered, especially in DMC patients who develop LBBB with QRS duration ≥ 150 ms and who have LVEF ≤ 35% and are symptomatic with standard medical therapy alone [67,89].

## 4. Risk of Surgery

The risk of dilated heart myocardiopathy is different in different stages of heart failure due to different conditions. Therefore, the stage and severity of HF should be considered when assessing perioperative risk [90].

### 4.1. Asymptomatic Left Ventricular Dysfunction

Most asymptomatic patients have decreased left ventricular diastolic function, but some asymptomatic patients also have decreased left ventricular systolic function. A study [91] of perioperative major adverse cardiovascular events (MACEs, which includes recurrent angina pectoris, acute myocardial infarction, severe arrhythmia, heart failure, and death from coronary heart disease) risk in vascular surgery showed that in high-risk procedures (open vascular surgery), the 30-day risk of MACEs was 14% in patients with normal left ventricular function, 23% in patients with asymptomatic isolated left ventricular diastolic dysfunction, and 31% in patients with asymptomatic left ventricular systolic dysfunction. The rate was up to 54% in patients with symptomatic heart failure.

### 4.2. Asymptomatic Heart Failure

Patients with a prior diagnosis of symptomatic HF, even when asymptomatic before surgery, are at increased risk after surgery [92]. The odd ratio for postoperative mortality was 1.53 with a 95% confidence interval from 1.44 to 1.63.

### 4.3. Symptomatic Heart Failure

Patients with symptomatic heart failure usually indicate an acute or terminal stage of the underlying disease. With decreased LVEF, the risk of MACEs increases progressively. For patients with symptomatic heart failure, the odds ratio for postoperative mortality was as high as 2.37 [92]. For patients with acute symptomatic heart failure, attention should be paid to controlling the symptoms, and the underlying disease should be treated actively. Therefore, surgery should be postponed whenever possible until the symptoms are improved and the patient’s status is stabilized [93]. However, as the disease progresses to the end stage, with the appearance of various complications, the patient’s condition becomes difficult to control or improve. At this time, surgical procedures should not be considered unless they are life-saving [94].

## 5. Timing of Surgery [90]

### 5.1. Emergent Surgery

Emergent surgery requires a surgical procedure completed within a limited time to save a patient’s life, limbs, or vital organs. However, a short period of time means that both assessment and optimization of the patient’s physiological state are rather limited, and there could be loopholes in the formulated anesthesia plan, which eventually leads to an increased risk for adverse events.

### 5.2. Scheduled or Elective Surgery

Operation as planned without inspection or intervention:(1)Low-risk scheduled or elective procedures for asymptomatic patients with heart failure or those at potential risk for heart failure (the risk of perioperative MACEs is <1%).(2)Intermediate-risk scheduled or elective surgery in patients with stable heart failure and good left ventricular functional reserve (the risk of perioperative MACEs is >1%).(3)Scheduled procedures for patients with stable heart failure and reduced left ventricular functional reserve, and a risk-benefit analysis should be performed (the risk of perioperative MACE is >1%).(4)A risk-benefit assessment should also be performed for scheduled surgery in patients with decompensated heart failure or heart failure with new-onset reduced ejection fraction (HFrEF). If surgery is inevitable, preoperative status should be optimized as much as possible, and intensive intraoperative monitoring should be performed.

## 6. Anesthetic Management

The points of anesthesia management in DCM are [95,96]:Avoid fluid overload.Avoid sudden increases in afterload and prevent arrhythmia or thromboembolic events.Maintain adequate myocardial contractility and ejection fraction to meet the requirement of perfusion to vital organs, especially the coronary arteries.

### 6.1. Intraoperative Monitoring

Appropriate monitoring is required to maintain optimal hemodynamics. ASA standard monitoring is necessary for all patients, including pulse oxygen saturation, noninvasive blood pressure, and five-lead electrocardiogram (ECG). Depending on a patient’s condition and the type of surgery, invasive arterial pressure monitoring, central venous pressure monitoring, neuromuscular monitoring, or transesophageal echocardiography (TEE) should be considered.

### 6.2. Choice of Anesthesia Technique

For DCM patients, there is no individual anesthetic technique that is superior to another. The choice depends on the anesthesiologists’ preference, the resources available, the patient’s condition, and the type of surgery. The benefits and issues of different anesthetic techniques are discussed below.

It is generally accepted that neuraxial anesthesia could be considered in DCM patients receiving surgeries on the lower abdomen or lower extremities. However, hypotension following epidural or spinal anesthesia is the primary concern for neuraxial anesthesia. Currently, there is no single vasopressor recommended for preventing or treating hypotension after neuraxial anesthesia in DCM patients, especially following spinal anesthesia [97,98]. Therefore, common α-adrenergic receptor agonists, such as phenylephrine, norepinephrine, or ephedrine, can all be used. One alternative to minimize the risk of hypotension after spinal anesthesia is to use titrated doses of local anesthetics. A comparison of continuous spinal anesthesia with titrated administration to single-dose spinal anesthesia revealed that the former technique resulted in a lower frequency of decreases in mean arterial pressure and significantly reduced ephedrine use [99]. Combined spinal epidural (CSE) anesthesia with low-dose spinal block and extension with an epidural top-up to extend the area and duration of blocking could also be considered for DCM patients. Currently, the CSE technique is more popular and widely used than the continuous epidural/spinal technique [100].

Unlike the potential failure block associated with spinal or epidural anesthesia, general anesthesia always provides excellent muscle relaxation and analgesia. The major concern with general anesthesia is that almost all general anesthetics could lead to myocardial depression (discussed below), which should be avoided in DCM patients. In addition, mechanical ventilation results in decreased venous return and cardiac output. Therefore, tidal volume can start at 6–8 mL/kg and be titrated to minimize its effect on cardiac filling.

### 6.3. Selection and Use of Drugs

DCM patients are extremely sensitive to the potential cardio-depressant effect associated with anesthetics. Unfortunately, cardio-depression has been reported with most, if not all, common general anesthetics. As myocardial depression caused by general anesthetics is dose-dependent, the dose, but not the type, of general anesthetics should be the primary concern.

Fentanyl produces little myocardial depression even at the dose of 30 μg/kg [101], which is the preferred choice of opioids for DCM patients. However, a large dose of fentanyl is associated with respiratory depression and delayed extubation. Midazolam is a commonly used sedative agent as it produces little myocardial depression or vasodilation [102]. Dexmedetomidine is another choice of preferred sedative agents due to its opioid-sparing effect. The major concern with dexmedetomidine is that it causes bradycardia and a resultant decrease in cardiac output [103,104]. Propofol, the most commonly used intravenous anesthetic, reduces left ventricular preload and afterload and produces myocardial depression [105]. Therefore, etomidate is advocated as an alternative to propofol as an induction agent for DCM patients with cardiac dysfunction due to its minimal cardiovascular effect. The direct effect of ketamine on the heart is negative inotropy, especially in heart failure [106]. However, when the central nervous system and autonomic nervous system are perfect, ketamine will produce a completely different response [107]. The effects of ketamine on hemodynamics are mainly reflected in sympathomimetic effects, including increasing cardiac output, heart rate, and arterial blood pressure. Some authors also prefer ketamine as an induction agent due to its positive inotropic effect [108].

Contemporary volatile anesthetics, i.e., isoflurane and sevoflurane, have been reported in DCM patients [109,110,111]. A previous study [112] showed that isoflurane does not have a beneficial effect on left ventricular afterload, and in the case of left ventricular dysfunction, left ventricular preload and reduced myocardial contractility are the main hemodynamic consequences of 1.1 and 1.5 MAC isoflurane. Sevoflurane does not sensitize the heart to epinephrine or cause alterations in hematologic or serum clinical chemistry values after repeated exposures in males after a single exposure [113]. Due to its low blood–gas partition coefficient and little airway stimulation, sevoflurane is preferred for rapid induction and emergence from anesthesia [114]. However, 8% sevoflurane was reported to lead to a second-degree atrioventricular block in a 12-year-old healthy female [115]. In summary, volatile anesthetics are generally used in low concentrations (up to 1 MAC) to minimize their undesired cardiovascular effects, such as vasodilation or myocardial depression.

### 6.4. Fluid Management

Excessive fluid leads to volume overload, pulmonary edema, and even heart failure [96]. However, too little fluid may result in insufficient ventricular filling and significantly reduced cardiac output for vital organ perfusion. Therefore, the appropriate amount of fluid infusion needs to be determined based on multiple parameters and a patient’s response. For example, arterial blood pressure, central venous pressure (CVP), HR, hemoglobin concentration, the level of lactic acid level and urine volume can be considered indicators to assess the amount of fluid infusion. Transesophageal echocardiography (TEE) can be used to continuously and directly monitor cardiac function and fluid status [116].

### 6.5. Postoperative Analgesia

Sufficient postoperative analgesia is key to the postoperative recovery of DCM patients, as excessive pain can lead to hemodynamic instability. Multimodal analgesia, which refers to nonsteroidal anti-inflammatory drugs, nerve blocks, and opioids, is performed to provide the desired analgesic effect [117].

## 7. Conclusions

This review summarizes the clinical features of patients with dilated cardiomyopathy and the key aspect of perioperative assessment and management of DCM patients undergoing noncardiac surgery. Optimal surgical outcomes depend on a full understanding of a patient’s pathophysiology and optimization of preoperative cardiovascular status. Afterload reduction, optimal preload, and minimal myocardial depression should be the goal of anesthetic management.

## Data Availability

All data are presented in the present manuscript.

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
