# Peer review of "Anesthetic Management of Patients with Dilated Cardiomyopathy Undergoing Noncardiac Surgery"

_medicina, 2023, doi:10.3390/medicina59091567_

Round 1

Reviewer 1 Report

The review is very preliminary & further details study is required to achieve the objectives which details are as follows:

1.      Author should mention the methodology of review, suggested to follow PRISMA, it is known for evidence-based minimum set of items for reporting in reviews and meta-analyses.

2.      Abstract should be more specific, focusing about the significance/benefits of review.

3.      Provide a brief about Dilated Cardiomyopathy (DCM) and comorbidity relationship, if any with related references.

4.      To attract more medical/ & nonmedical readers, author is suggested to provide model figures/flowcharts of Dilated Cardiomyopathy (DCM) and related mechanism.

5.      Authors are suggested to focus more about genetic predisposition, physiochemical behavior and environmental players (activators/repressor) in context with DCM Dilated Cardiomyopathy (DCM) in introduction part, because these are a vital for the said case.

Minor correction is required.

Author Response

  1. Author should mention the methodology of review, suggested to follow PRISMA, it is known for evidence-based minimum set of items for reporting in reviews and meta-analyses.

REPLY: Thanks for reminding us on this. But we have to admitted that our review is summary of recent advances in perioperative management of DCM patients, but not a quantitative meta-analysis. Therefore, we don’t think the PRISMA is suitable for us. But if you insist, we are happy to include it in our next revision.

  1. Abstract should be more specific, focusing about the significance/benefits of review.

REPLY: Thanks very much for reminding us on this important issue. We have revised this part, please see LM 13-17.

  1. Provide a brief about Dilated Cardiomyopathy (DCM) and comorbidity relationship, if any with related references.

REPLY: Thanks very much for this interesting comment!  But there is little literature talking about DCM and comorbidity. Rather, DCM is closely associated with reduced LVEF and sudden cardiac death. Please see our discussion in LM 64-73.

  1. To attract more medical/ & nonmedical readers, author is suggested to provide model figures/flowcharts of Dilated Cardiomyopathy (DCM) and related mechanism.

REPLY: Thank you for your suggestions! Due to the very limited time, we are unable to provide a  flow chart. We are happy to provide it in our next revision if you think it is necessary.

  1. Authors are suggested to focus more about genetic predisposition, physiochemical behavior and environmental players (activators/repressor) in context with DCM Dilated Cardiomyopathy (DCM) in introduction part, because these are a vital for the said case.

REPLY: Thanks very much for this interesting comment! Regarding the content you mentioned, we have added the effect of environment players, please refer to LM 48-50, as well as reference no. 13.

Reviewer 2 Report

The authors described the key elements of anesthetic management of dilated cardiomyopathy. The article is well written and provides practical useful information. I have only few suggestions:

Line 21: replace contraction function with contractility

Line 27: add more recent citation

Line 46: add more recent citation

Line 50: the section preoperative assessment could be enriched with a table summarizing risk factors and therapeutic goals

Line 70-74: rephrase the sentence

Finally, as DCM patients often have symptomatic heart failure with frequent worsening episodes I suggest authors to briefly discuss in preoperative assessment:

- timing of surgery

- hemoglobin levels

- cardiac implantable electronic devices management

Minor English editing required

Author Response

(The authors gave the same response as above.)

Reviewer 3 Report

The Authors described all main clinical and echocardiographic characteristics of dilated cardiomyopathy (DCM), as well as the preoperative assessment of DCM patients, with particular focus on anesthetic management.

I think that the review is well written and very interesting.

I have only a suggestion with regards to echocardiographic evaluation of DCM patients.

Given that in the paragraph 3. Preoperative Assessment the Authors stated that echocardiography may allow to detect the presence of LV thrombus, after line 78, the Authors could also specify that recent evidences suggest that pulsed wave (PW) Tissue Doppler Imaging (TDI) may provide a rapid characterization of the left ventricular (LV) thrombotic mass mobility. Notably, the peak anterograde PW-TDI velocity, as recorded at the free mobile portion of the LV thrombus, could be a marker of the driving force for thrombus embolization (Please cite the following references: PMID: 33231099 and PMID: 34395561).

Author Response

I have only a suggestion with regards to echocardiographic evaluation of DCM patients.

Given that in the paragraph 3. Preoperative Assessment the Authors stated that echocardiography may allow to detect the presence of LV thrombus, after line 78, the Authors could also specify that recent evidences suggest that pulsed wave (PW) Tissue Doppler Imaging (TDI) may provide a rapid characterization of the left ventricular (LV) thrombotic mass mobility. Notably, the peak anterograde PW-TDI velocity, as recorded at the free mobile portion of the LV thrombus, could be a marker of the driving force for thrombus embolization (Please cite the following references: PMID: 33231099 and PMID: 34395561).

REPLY: Thanks very much for this insight comment! According to your suggestion, we have added relevant content to the article,LM 128-148. And the paper you mentioned is also cited in our revision.

Round 2

Reviewer 1 Report

All necessary corrections have been done. It can be approved.